# Endophytic Bacteria Colonizing the Petiole of the Desert Plant *Zygophyllum dumosum* Boiss: Possible Role in Mitigating Stress

**DOI:** 10.3390/plants11040484

**Published:** 2022-02-11

**Authors:** Jansirani Srinivasan, Janardan Khadka, Nurit Novoplansky, Osnat Gillor, Gideon Grafi

**Affiliations:** 1French Associates Institute for Agriculture and Biotechnology of Drylands, Jacob Blaustein Institutes for Desert Research, Ben-Gurion University of the Negev, Midreshet Ben-Gurion 84990, Israel; jansirani085@gmail.com (J.S.); janardankhadka@gmail.com (J.K.); nuritnov@bgu.ac.il (N.N.); 2Zuckerberg Institute for Water Research, Jacob Blaustein Institutes for Desert Research, Ben-Gurion University of the Negev, Midreshet Ben-Gurion 84990, Israel; gilloro@bgu.ac.il

**Keywords:** desert plants, phyllosphere endophytic bacteria, *Zygophyllum dumosum*, Actinobacteria, Firmicutes, 16S rRNA genes, endophyte metabolites, stress-related metabolites

## Abstract

*Zygophyllum dumosum* is a dominant shrub in the Negev Desert whose survival is accomplished by multiple mechanisms including abscission of leaflets to reduce whole plant transpiration while leaving the fleshy, wax-covered petioles alive but dormant during the dry season. Petioles that can survive for two full growing seasons maintain cell component integrity and resume metabolic activity at the beginning of the winter. This remarkable survival prompted us to investigate endophytic bacteria colonizing the internal tissues of the petiole and assess their role in stress tolerance. Twenty-one distinct endophytes were isolated by culturing from surface-sterile petioles and identified by sequencing of the 16S rDNA. Sequence alignments and the phylogenetic tree clustered the isolated endophytes into two phyla, Firmicutes and Actinobacteria. Most isolated endophytes displayed a relatively slow growth on nutrient agar, which was accelerated by adding petiole extracts. Metabolic analysis of selected endophytes showed several common metabolites whose level is affected by petiole extract in a species-dependent manner including phosphoric acid, pyroglutamic acid, and glutamic acid. Other metabolites appear to be endophyte-specific metabolites, such as proline and trehalose, which were implicated in stress tolerance. These results demonstrate the existence of multiple endophytic bacteria colonizing *Z. dumosum* petioles with the potential role in maintaining cell integrity and functionality via synthesis of multiple beneficial metabolites that mitigate stress and contribute to stress tolerance.

## 1. Introduction

Plants are sessile organisms that are constantly exposed to biotic and abiotic stresses. Consequently, plant have evolved a plethora of mechanisms including morphological, chemical, biochemical, molecular, and developmental mechanisms to enable them to cope with their ever-changing environment. This is well demonstrated in desert plants, such as *Zygophyllum dumosum* Boiss, that rely on multiple mechanisms to survive the vulnerable desert environment [1]. It is a dominant shrub on the rocky limestone southeast-facing slopes of the Negev Desert and is well adapted for growing in the harsh desert environment and can tolerate drought and the high salt conditions that commonly prevail in the desert ecosystem [2,3]. The plant is highly active during the winter producing new leaves and flowers (Figure 1A). The leaf is compound and consist of a pair of fleshy leaflets, which are carried on a thick, fleshy, and wax-covered petiole (Figure 1B). It is semi-deciduous during the dry summer (Figure 1C); that is, it sheds its leaflets, while leaving the thick, fleshy petiole alive but dormant during the dry season (Figure 1D) [4]. At the beginning of the winter, metabolic processes reactivate in petioles and growth resumes, culminating in production of new stems carrying new compound leaves from buds located at petiole axils [1,5]. Entry into a dormant state probably represents the major mechanism underlying petiole survival and the maintenance of cell components’ integrity during the dry season, which is accompanied by genome compaction and protein synthesis arrest [4,5]. How these petioles survive the long, hot summer while keeping the integrity of cellular components is still an open question. Thus, we hypothesized that in addition to morphological, biochemical, molecular, and developmental mechanisms, *Z. dumosum* plants might be associated with endophytic bacteria to increase their success in the desert ecosystem.

Endophytes are bacteria or fungi that inhabit the internal tissues of plants displaying synergistic or commensal mode of interactions and appear to be essential components of the plant and instrumental in maintaining plant health, growth, and development, and in facilitating plant responses to environmental stresses [6,7]. A wide range of bacteria and fungi were isolated from internal tissues of a variety of plant species. These include the most common endophytes isolated from phyllosphere, namely, *Bacillus* species as well as species of *Arthrobacter, Fictibacillus,* and *Kocuria* [8,9,10,11,12]. Some endophytes promote plant growth by making nutrients available for absorption by plants [13,14], providing anti-pathogenic substances [15,16], and producing plant phytohormones and substances that mitigate abiotic stresses [17,18].

Studies describing desert plants associated endophytes mostly focused on the rhizosphere [19,20,21]. Indeed, it has been demonstrated that *Z. dumosum* rhizosphere is associated with bacteria that were shown to confer stress protection [19]. Specifically, the bacterial isolate, *Dietzia cinnamea*, was shown to augment desiccation tolerance and significantly enhance the growth and yield of corn and wheat in lab and field trials [22,23]. Recent studies addressing the occurrence of desert endophytes in above-ground tissues revealed phylogenetically diverse communities dominated by Proteobacteria, Actinobacteria, and Firmicutes [24,25,26]. Additionally, surveys of perennials in the Atacama and Saudi Arabian deserts revealed highly regional and species-specific endophytes [27,28].

Here, we investigated the occurrence of endophytes in the petiole, the leaf part that persists during the dry season, of the desert plant *Z. dumosum* (Figure 1D). We hypothesized that endophytes colonizing the petiole might have the capacity to relieve the stress conditions inflicted by prolonged drought and heat, assist in keeping the integrity and functionality of cells during the dry season. and consequently contribute to plant survival in the desert environment.

## 2. Materials and Methods

### 2.1. Petiole Collection for Isolation of Endophytes and Preparation of Extracts

Petiole samples for endophytes isolation were collected from *Zygophyllum dumosum* shrubs growing in Central Negev Highlands (30°51.12′01″ N 34°46.06′42″ E, at 511 m elevation) in February 2020 from 12 plants within the 3–7 m distance between them. Samples were kept on ice during collection in the field and processed the same day for isolation of endophytes.

For preparation of petiole extracts (PEs), we used petioles that were collected during the summer (PE-S) and winter (PE-W) of 2010 and kept frozen at −80 °C until used [4]. Defrosted petioles (1 g) were first washed with sterile water to remove dust and ground with 10 mL of 20 mM of MES (2-(N-morpholino)-ethanesulfonic acid) buffer (pH 6) by using a mortar and pestle. Homogenates were filtered through 100 µm mesh filter, centrifuged at 10,000 rpm for 5 min, and sterilized by further filtering through 0.22 µm spin filter. Petiole extracts were kept frozen at −80 °C until used.

### 2.2. Isolation and Identification of Endophytic Bacteria from Petioles

Fresh petioles (2 g) were surface sterilized using 70% ethanol for 1 min, followed by 2.5% sodium hypochlorite for 4 min and 70% ethanol for 30 s, and rinsed twice with phosphate-buffered saline (PBS). Surface sterile petioles (SS-petioles) were ground with 25 mL PBS using a sterile mortar and pestle. Extracts were incubated at 25 °C for 3 h at 100 rpm followed by centrifugation (400 rpm, 5 min). Supernatant was collected and 100 µL were spread onto nutrient agar (HiMedia, Mumbai, India) plates. Plates were incubated at 27 °C for 4 to 10 days, and then colonies were isolated into fresh media. Out of 76 isolates, 60 were selected for DNA extraction and sequence analysis. DNA was extracted from each isolate by lysis (lysis buffer, Qiagen), and the extracted DNA was stored at −20 °C until used. The extracted DNA (50 ng) was used as a template for 16S rRNA gene amplification by PCR using 200 nM each primer of 27-F 5′-AGAGTTTGATCMTGGCTCAG and 1492-R 5′-TACGGYTACCTTGTTAC GACTT [29], 2.5 mM MgCl_2_, 0.8 μL of DreamTaq DNA polymerase (5 units/μL, Thermo Fisher, Waltham, MA, USA), 5 μL 10× DreamTaq buffer, 5 μg of bovine serum albumin (New England Biolabs, Ipswich, MA, USA), and 0.2 mM of dNTPs (TaKara, Kusatsu, Shiga, Japan). Reaction conditions were as follows: 95 °C for 2 min; then 32 cycles of 95 °C for 30 s, 54 °C for 30 s, and 72 °C for 90 s; and culminating with 72 °C extension for 5 min. PCR products were visualized on agarose gel and the amplicons were sequenced in HyLabs facility (Rehovot, Israel). The resulting sequences (Appendix A) were BLAST analyzed against the NCBI database. Phylogenetic tree analysis was done using MEGA X software (https://www.megasoftware.net/, accessed on 16 July 2020). Isolated endophytes were stored at −80 °C as glycerol stocks.

### 2.3. Bacterial Growth in the Presence of Petiole Extracts (PEs)

Single colonies of selected isolates were inoculated in 5 mL nutrient broth (NB; HiMedia, Mumbai, India) for 24 h at 29 °C with 180 rpm constant shaking. The culture was diluted 1:10 in fresh NB or in fresh NB supplemented with 10% of PE-S or PE-W (and incubated at 29 °C in a plate reader (TECAN, Männedorf, Switzerland) for 23–30 h. Bacterial growth was monitored at 6 h intervals. Statistical analysis of bacterial growth was performed by unpaired *t*-test with GraphPad software. Asterisks indicate statistical significance (*p* < 0.001).

### 2.4. Analysis of Primary Metabolites from Bacteria

Three bacteria were selected: *Arthrobactor agilis*, *Kocuria rosea,* and *Bacillus frigoritolerans*, which represent both phyla and display different mode of growth in the presence of petiole extracts. To test their metabolomes, the bacterial strains were grown in either NB or NB supplemented with PE-S (at 1:10 dilution). Upon reaching the stationary phase, the isolates were pelleted by centrifugation at 1000 rpm for 10 min, washed twice in double distilled water (DDW), and lyophilized. A precooled mix of 25 µL of methanol (Merck) and 10 µL of chloroform (Merck) in 10 mL of Milli-Q ultrapure water was used to sort the samples. Then, 200 µL of Ribitol (Cortecnet Corporation, Mill Valley, CA, USA) and sorbitol (Merck) were diluted in the precooled mix to be used as the internal standard (1 mg mL^−1^). Next, 500 µL of the precooled mix was added to the lyophilized bacteria samples, vortexed, and 50 µL precooled methanol was added. The resulting solution was vortexed for 10 min at 25 °C on an orbital shaker (New Brunswick Scientific, Edison, NJ, USA), then sonicated for 10 min in an ultra-sonication bath (Thomas Scientific, Swedesboro, NJ, USA) at 25 °C. The samples were centrifuged at 13,000 rpm for 10 min, the supernatant was transferred to a new tube, and 75 µL of chloroform (Merck) was added. The resulting solution was vortexed for 10 s and centrifuged the for 5 min at 13,000 rpm. The supernatant was transferred to a fresh tube and dried in a SpeedVac (Thermo Fisher, Waltham, MA, USA).

For the GC-MS analysis, 40 µL methoxyamine hydrochloride (Merck, Darmstadt, Germany) supplemented with 20 mg mL^−1^ pyridine was added to the dried samples. The resulting solutions were incubated for 2 h at 37 °C and 100 rpm, then 70 µL MSTFA reagent (*N*-Methyl-*N*-trimethylsilylfluoroacetamide; Merck, Darmstadt, Germany) and 7 µL of alkane mix (Merck, Darmstadt, Germany) were added, and the resulting solutions were incubated at 37 °C for 30 min at 100 rpm. The samples were analyzed with gas chromatography–mass spectrometry (GC-MS) (Agilent, Santa Clara, CA, USA) as previously described [30].

For the GC-MS data analysis, to analyze the unknown spectrum, the MassHunter Qualitative, Unknown, and Quantitative analyses (Agilent B.07.00, 10) was used against RI libraries downloadable from the Max-Planck Institute for Plant Physiology in Golm (http://gmd.mpimp-golm.mpg.de/, accessed on 16 August 2020). Metabolite relative abundance was determined by normalizing the intensity of the peak of each metabolite to the ribitol internal standard. The results were visualized in heatmaps and PCA plotted using MetaboAnalyst webtool (https://www.metaboanalyst.ca/home.xhtml, accessed on 16 August 2020).

### 2.5. Nutrient Analysis

First, 1 g of summer and winter petioles were washed and homogenated with 1 mL of DDW. Homogenates were passed through 100 μM filter mesh and centrifuged at high speed (10,000 rpm). The supernatant was collected after being filtered through 0.22 µm spin filter, and 200 μL of each sample were diluted with 5.8 mL of Milli-Q water and subjected to unbiased nutrient analysis by the inductively coupled plasma-optical emission spectroscopy (ICP-OES) using ICP-720-ES (Varian Inc., Palo Alto, CA, USA).

### 2.6. Statistical Analysis

Statistical analyses were performed by one-way ANOVA calculator, with Tukey test calculator (https://www.stepbystepsolutioncreator.com/st/anova, accessed on 16 August 2020). Unpaired *t*-test was performed using the GraphPad QuickCalcs Web site: https://www.graphpad.com/quickcalcs/ttest1/?Format=C (accessed on 16 August 2020). All assays were repeated at least three times and representative results are shown.

## 3. Results

### 3.1. Cultivation of Endophytic Bacteria from Z. dumosum Petioles

Endophytic bacteria were isolated from surface-sterile petioles of the desert plant *Z. dumosum* and cultured on nutrient agar plates. Some of the isolated bacteria produced colored pigments and different morphology (Figure 2A). Out of 76 isolates, 60 were selected for 16S rDNA PCR amplification and sequencing. Based on sequence alignment of the 16S rDNA, a phylogenetic tree was generated clustering the isolated bacteria into two phyla, Firmicutes and Actinobacteria (Figure 2B). BLAST analysis of the resulting sequences against the ‘nr’ nucleotide database in NCBI revealed 21 distinct bacteria (Table 1). Many of the identified species have been reported previously to colonize internal tissues of various plant species, such as *Arthrobacter agilis*, *Dietzia lutea*, and *Bacillus frigoritolerans*, further supporting that the bacterial isolates might be endophytes. Notably, we could not detect endophytic fungi under the culturing medium (nutrient agar) and the procedure applied in the present work.

### 3.2. Z. dumosum Petiole Extracts (PEs) Promote Endophytic Bacterial Growth

To gain insight into the mode of the plant–microbe interaction, we sought to examine the potential contribution of *Z. dumusum* PE to endophytes growth. We observed that endophytes are generally growing slowly in NB medium; therefore, we examined the capacity of plant extracts derived from petioles collected during the summer (PE-S) and the winter (PE-W) to control endophyte growth. To this end, selected endophytes from both phyla were grown in NB medium only or in NB medium supplemented with PE-S or PE-W. Measurements of the optical density (OD_600_) were taken every 6 h in a course of 23–30 h. Results showed that essentially all endophytes displayed significant increase in growth rate in the presence of summer or winter PEs (Figure 3). No significant difference could be found in most endophytes’ growth under summer and winter petiole extracts, except for *Georgenia satyanarayana,* whose growth was significantly enhanced under PE-S compared to PE-W. Notably, *A. agilis* repeatedly displayed a peculiar slow, bi-phasic-like growth curve in both NB and NB + PEs medium.

### 3.3. Metabolic Analysis of Endophytic Bacteria

To expand our understanding of the endophyte–plant interaction, we analyzed the primary metabolite profiles of selected endophytes when grown with or without PE-S. For primary metabolite analysis, we selected three endophytes, namely, *K. rosea* and *B.*
*frigoritolerans* (representing both phyla), as well as *A. agilis,* because of its unique slow, biphasic-like growth on bacterial growth medium (NB). We identified 67 primary metabolites (Appendix A). A principal component analysis (PCA) showed that the addition of PE-S to bacterial growth medium (NB) had a different effect on the primary metabolite profile of each endophyte. Accordingly, the first principal component (PC 1) demonstrates 92% and 90.1% of the variance for *K. rosea* and *B.*
*frigoritolerans,* respectively, separating between the bacteria grown in NB and the bacteria grown in NB + PE (Figure 4A). However, the effect of PE on *A. agilis* metabolite profile was marginal and not significant inasmuch as PC 1 clustered treatments together, while PC 2 demonstrating 30.8% of the variance clearly separates the treatments. The heatmap generated for bacterial primary metabolites showed that PE affected primary metabolites of each bacteria differently, demonstrating that plant–endophyte interaction is a species-specific process (Figure 4B).

The relative content of certain metabolites identified in each endophyte grown on NB or in the presence of PE is shown in Figure 5. This demonstrates common metabolites that are synthesized at relatively high levels by all endophytes, namely phosphoric acid, pyroglutamic acid, glutamic acid, alanine, and glycine. Other metabolites synthesized at high levels appear to be endophyte-specific metabolites. Accordingly, trehalose is produced at high levels in *K. rosea* and *A. agilis* but at a very low level in *B. frigoritolerans*. On the other hand, the synthesis of the monosaccharide lyxose is increased in *B. frigoritolerans* under NB + PE, while proline is increased in *K. rosea* under NB + PE. Notably, while in *K. rosea* metabolites are upregulated in the presence of PE, in *B. frigoritolerans* they are essentially downregulated, and no significant change is observed for *A. agilis*. Notably, proline and trehalose were implicated in plant response to stress, particularly in response to salinity where they act as osmoprotectants and antioxidants to increase stress tolerance [52,53,54].

### 3.4. Z. dumosum Petioles Have Increased Levels of Nutrients during the Dry Season

We assumed that the endophytes *K. rosea* and *A. agilis* produce a relatively high level of the osmoprotectants trehalose and proline in response to abiotic stress, most likely because of the high salt concentration they encounter in the internal tissues of the petiole throughout the year. We thus investigated the nutrient profiles of extracts from petioles collected during the wet and the dry seasons. Figure 6 shows that the level of most macro- and microelements was significantly elevated in petioles during the dry season. This increase in concentration of most elements examined might be related to the loss of water from petioles that occurs during the dry season. Sodium, calcium, and sulfur are among the major elements accumulated to high levels in *Zygophyllum* petioles regardless of the season and whose levels are significantly increased during the dry season (Figure 6A). Other macroelements whose levels were increased during the summer are bromine, potassium, and strontium. The level of several microelements was increase during the dry season including iron, zinc, manganese, and copper, the latter is the major element showing remarkable increase (2.8 fold) during the dry season (Figure 6B). Calculation of the sodium concentration in a single petiole showed that in summer petioles concentration is higher (123 mM) than in winter petioles (79 mM) (Appendix A). Thus, the internal chemical environment of the petiole is modified substantially on the transition from the wet to the dry season.

## 4. Discussion

Most studies on desert plant associated bacteria have focused on the rhizosphere [55], while the phyllosphere bacteria have gained less attention [56,57]. The present study addressed endophytes colonizing the internal tissue of *Z. dumosum* petioles and assessed their biological significance. Twenty-one different endophytes were isolated from *Z. dumosum* petioles. The 16S rRNA encoding gene sequence alignment and phylogeny analysis classified all isolated endophytes into two phyla, Firmicutes and Actinobacteria. This is consistent with previous reports demonstrating that phyllospheric bacteria are isolated from desert plants, predominantly belong to Firmicutes and Actinobacteria phyla [58,59], which are also well represented in the microbial community of desert dust [60]. Among the 21 isolates, 10 bacteria were previously reported as endophytic bacteria (either rhizospheric or phyllospheric) including *A. agilis, B. megaterium,* and *B. halotolerans.* The other isolates include root-associated (rhizobacteria) and soil bacteria [12,43,50]. Several isolates were not reported previously as endophytic bacteria, including *G. satyanarayanai*, *S. xinjiangense*, *B. licheniformis*, *D. robiginosus*, and *P. basanitobsidens*, and may be specific to the host. Indeed, desert plants have been reported to contain endophytic bacteria whose composition is affected by the host plant [28].

In plant–microbe symbiosis, plant and bacteria support each other and thus increase their survival in their specific ecological niche. Accordingly, isolated endophytic bacteria grew slowly in a defined culture medium, but their growth was significantly induced by *Z. dumosum* PEs; the nature of bacterial growth promotion by PEs is presently unknown. These results are consistent with a previous report demonstrating regrowth of endophytic bacteria isolated from *Cucurbita pepo* with *C. pepo* plant extracts [61]. Notably, although nutrients and particularly sodium accumulated to high levels in summer petioles, we calculated a relatively low, negligible concentration of sodium in the growth medium supplemented with petiole extracts (0.65 mM for winter and 0.95 mM for summer petiole).

Many endophytic bacteria isolated from a variety of plant species were reported to support plant growth by various means including production of phytohormones, siderophores, and other substances that activate plant stress response and confer tolerance to biotic and abiotic stresses [62,63,64,65,66,67]. Some of the beneficial substances produced by endophytes were identified including antimicrobial substances, such as oomycin A and taxol, which protect plants from potential pathogens [68,69], or growth factors, such as phytohormones auxin, gibberellic acid, and siderophores, or iron-chelating compounds that assist in transport iron across cell membranes [70,71].

The isolated endophytes often produced colored colonies on NB medium including orange, yellow, green, and pink colonies (Figure 2A). Additionally, strong scent was emitted from *D. lutea* and *B. frigoritolerans* colonies. Previous studies have reported that several plant-associated bacteria produce pigments that are used in paints, crimson ink, cosmetics, and food coloring as well as pigments that possess anticancer activity [72]. Bacterial pigments such as anthocyanins, carotenoids, riboflavin, and melanin may provide protection from environmental stresses such as UV radiation and oxidative conditions [72,73,74]. In *Serratia marcescens* bacteria, the red pigmentation is conferred by prodiginines, which have a wide range of biological properties including antibacterial, antifungal, and anticancer activities [75,76]. Phenazine pigments are exclusively produced by bacteria; they are water soluble and excreted by the bacteria; phenazines possess antibiotic activities against other bacteria, fungi, or plant/animal tissues [77,78]. Aquatic cyanobacteria produce yellow-green pigmentation conferred by the secondary metabolite, scytonemin, which is formed when bacteria are exposed to sunlight as a protection means against the damaging UV light. Halophilic bacteria produce orange to red pigmentations, which were found to be carotenes [79,80], which are involved in multiple processes including photosynthesis and photoprotection as well as in ABA synthesis [81,82]. Hence, the pigments produced by endophytic bacteria might be useful in enhancing tolerance of *Z. dumosum* to biotic and abiotic stresses.

Some of the mechanisms underlying the beneficial effect of endophytes on plant response to stress might be related to specific metabolites produced by the endophytes. These metabolites may act directly or indirectly inducing plant to produce substances (metabolites and proteins) that mitigate the effects of stress. The metabolic analysis revealed several metabolites, which are relatively highly produced by the endophytes including alanine, phosphoric acid, glycine, aspartic acid, pyroglutamic acid, glutamic acid, and putrescine. Interestingly, each endophyte had a distinct response to the presence of PE. While these highly produced metabolites were upregulated in *K. rosea* in the presence of PE, they were downregulated in *B.*
*frigoritolerans* and unchanged in *A. agilis*. Certain metabolites appeared to be species-specific metabolites. Accordingly, *B.*
*frigoritolerans* produces a relatively high level of the monosaccharide lyxose, while *K. rosea* and *A. agilis* have a relatively very high level of the disaccharide trehalose. Some metabolites produced by the endophytes have been described previously and shown to confer stress tolerance. Thus, trehalose produced by endophytes may act as an osmoprotectant [83] and assist in keeping the integrity and functionality of cells during the dry season and, consequently, contribute to survival in the desert environments [84]. Notably, it is accumulated in high levels in leaves of resurrection plants [85,86]. Trehalose is prevalent in bacteria, fungi, and invertebrates, and it accumulates during heat shock and enhances thermotolerance, at least partly by reducing aggregation of denatured proteins [87,88].

Pyroglutamic acid is produced at a relatively high level by all the endophytes that we examined. It accumulates in response to salt stress and might act as an osmoprotectant [89]. Pyroglutamic acid is produced de novo in response to osmotic stress, probably by the enzyme glutamine synthetase, which cyclizes the glutamate into pyroglutamic acid in the absence of ammonia [54,90].

As multifunctional amino acid, proline is known to be upregulated in plants subjected to stresses such as drought and salt [91]. It might fulfil various roles under stress conditions, including scavenging reactive oxygen species, thus protecting cellular functions by stabilizing proteins, membranes, and subcellular structures [92,93,94]. Two of the isolated endophytes, *Bacillus megaterium* and *B. licheniformis*, were reported to induce drought tolerance in wheat under drought stress, at least partly by enhancing proline level [95]. The production of proline and trehalose by certain endophytes could be triggered by the relatively increased level of salt accumulated in petioles throughout the year as a means for increasing tolerance to salinity [96]. Indeed, calculation of sodium concentration within petioles shows a significant increase in summer petioles (~123 mM) compared to winter petioles (~79 mM) (Appendix A). Changes in the internal chemical environment and metabolic activities (e.g., protein synthesis, respiration, and photosynthesis) of petioles during the transition from the wet to the dry season [1,4] might have an effect on endophyte community composition [97].

Finally, the increased level of sulfur in summer petioles could have a function in mitigating abiotic stress conditions such as salinity and drought; sulfate-containing metabolites are significantly increased in *Z. dumosum* petioles during the dry season [98]. It is an essential macroelement in plants and is involved in shaping the plant microbiome [99] and in production of multiple compounds necessary for metabolic processes under optimal and stress conditions, including amino acids biosynthesis (cysteine, methionine) and S-adenosyl methionine (SAM), a universal donor for methyl group, which is also involved in epigenetic control of chromatin structure and function [100,101].

## 5. Conclusions

Endophytes in *Z. dumosum* petioles may act separately or in concert to mitigate the adverse effects of stresses that prevail in the desert ecosystem. The protective activities conferred by endophytes could be mediated by stress-related metabolites produced by the endophytes including pigments, proline, and trehalose. Alternatively, metabolites produced by endophytes may trigger petiole cells to produce beneficial substances (proteins and metabolites) that protect cells from hazardous stress conditions and maintain cellular integrity [102]. We anticipate changes in endophyte community composition during the transition from the wet and the dry season owing to changes in the internal chemical environment and metabolic activity of the petiole. Further work is necessary to assess endophyte community composition during the wet and the dry season and to explore the potential of other isolated endophytes to produce beneficial substances. These endophytes may be used for inoculating important crop plants to improve their performance under abiotic stress conditions, whose severity and frequency are expected to increase in the face of climate change.

## Figures and Tables

**Figure 1 plants-11-00484-f001:**
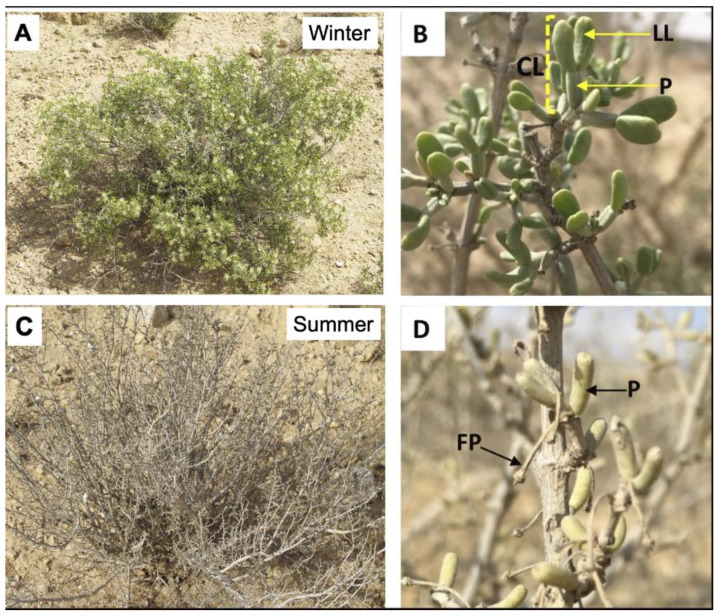
Phenology of *Zygophyllum dumosum* Boiss during the winter and summer of 2019 at Sede Boqer research area. (**A**) A typical appearance of *Z. dumosum* shrub during the wet season (winter). (**B**) Appearance of a *Z. dumosum* branch during the wet season carrying new compound leaves (CL) each consists of two leaflets (LL), which are carried on a thick and fleshy petiole (P). (**C**) *Z. dumosum* shrub appearance during the dry season (summer). (**D**) A *Z. dumosum* branch (dry season) carrying the remaining petioles. FP, flower pedicel.

**Figure 2 plants-11-00484-f002:**
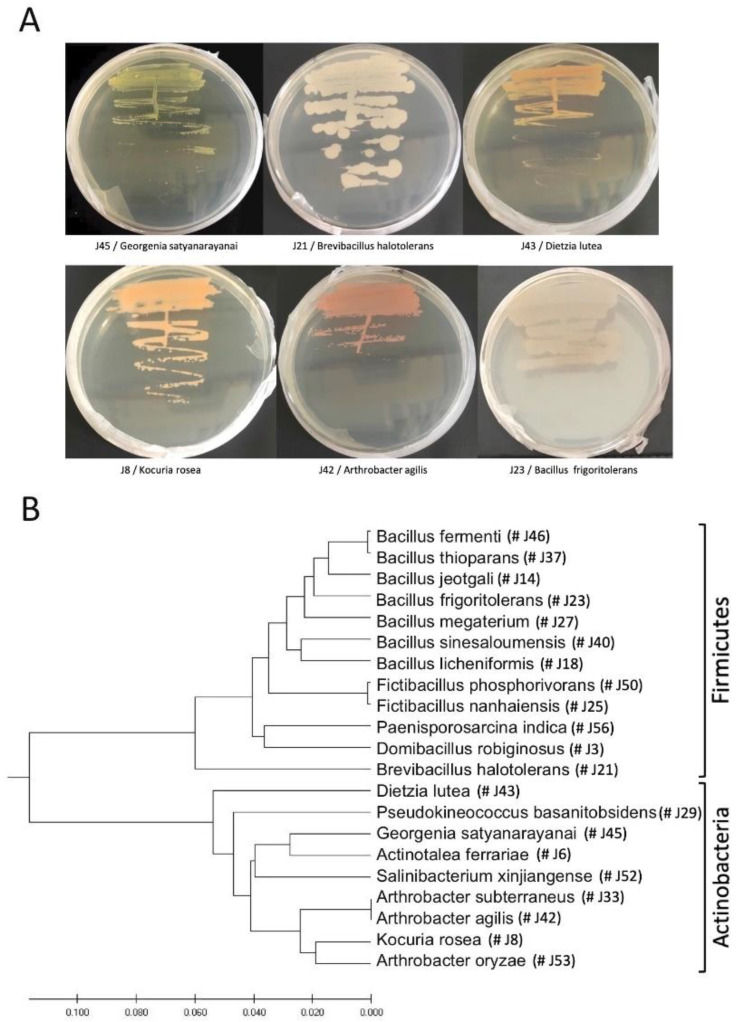
Isolation of endophytes from *Z. dumosum* petioles on nutrient agar plates. (**A**) Examples of endophytic bacteria producing colors. (**B**) Phylogenetic relationship of the endophytic bacteria isolated from petioles dividing the identified endophytes into two phyla, Firmicutes and Actinobacteria.

**Figure 3 plants-11-00484-f003:**
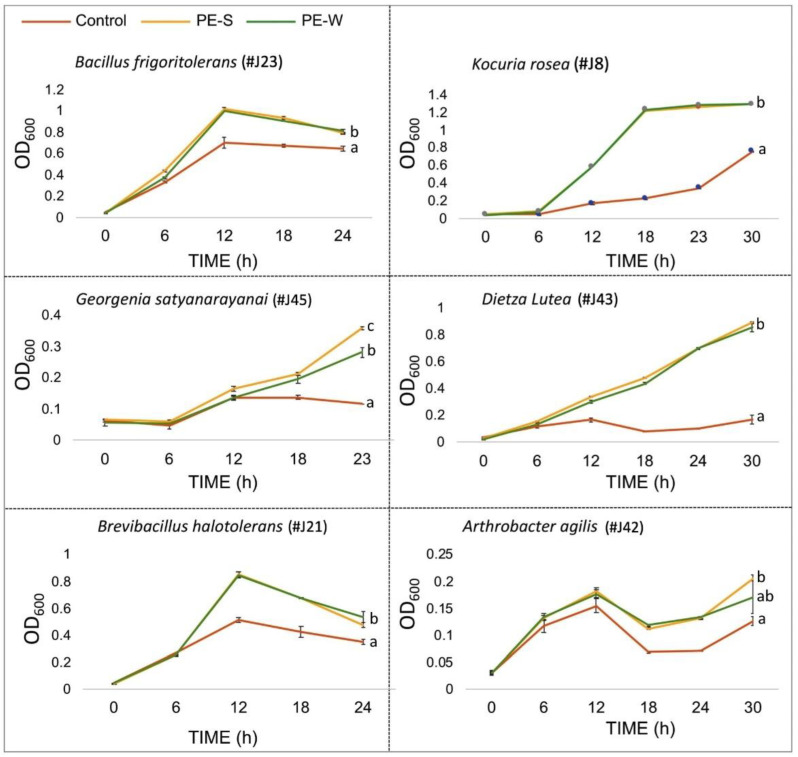
Petiole extracts (PEs) promote bacterial growth. The indicated bacteria were grown in a conical flask in NB (control, red line), or in NB supplemented with 10% of summer or winter PEs (PE-S, orange line; PE-W, green line). Bacterial growth was monitored by measuring the OD_600_ of the culture every 6 h in a course of 23–30 h. Each treatment was performed in triplicates, and error bars represent the standard deviation. Statistical significance between treatments was determined for each endophyte by one-way ANOVA with Tukey test calculator, and different letters indicate statistically significant differences between treatments (*p* < 0.01).

**Figure 4 plants-11-00484-f004:**
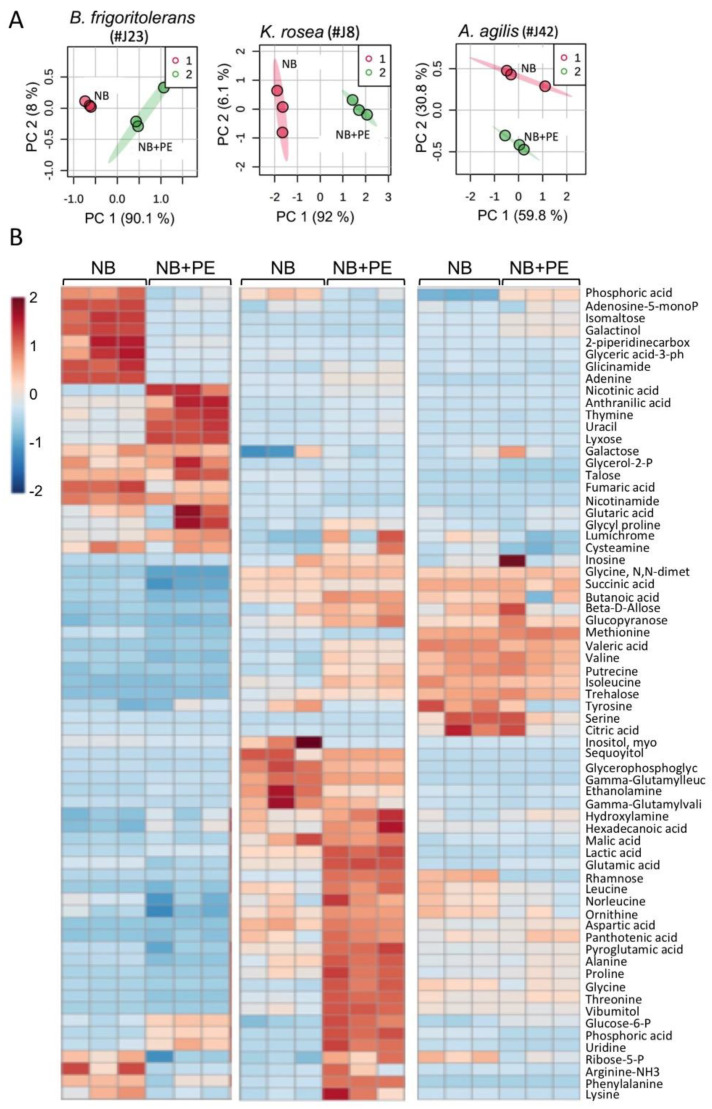
Analysis of primary metabolites synthesized by the isolated endophytes *K. rosea*, *B. frigoritolerans,* and *A. agilis* (**A**) Score plots of principal component analysis (PCA) comparing bacterial metabolites grown in NB versus NB supplemented with PE (NB + PE). (**B**) A heatmap demonstrating differential synthesis of primary metabolites by the indicated endophytic bacteria cultured in NB or in NB + PE. The color key represents the fold change (log_2_) of metabolites between NB and NB + PE samples.

**Figure 5 plants-11-00484-f005:**
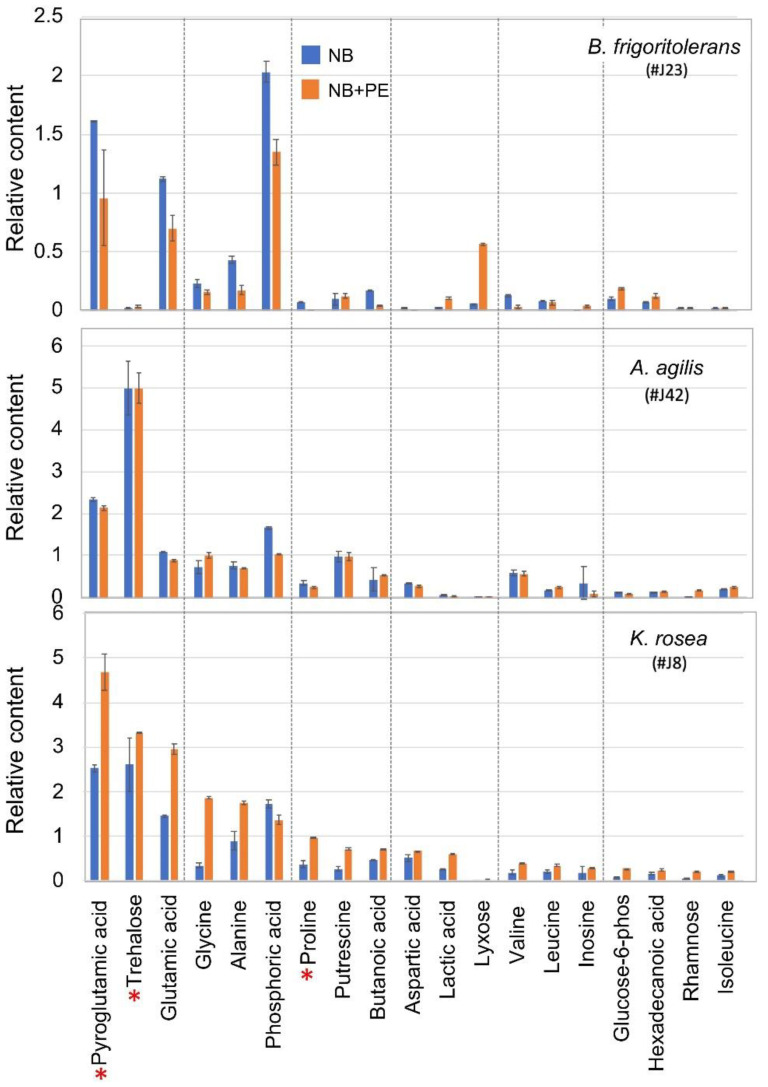
Relative content of certain metabolites synthesized in the indicated endophytic bacteria cultured in NB (blue) or in NB + PE (orange). Red asterisks indicate metabolites implicated in response to stress.

**Figure 6 plants-11-00484-f006:**
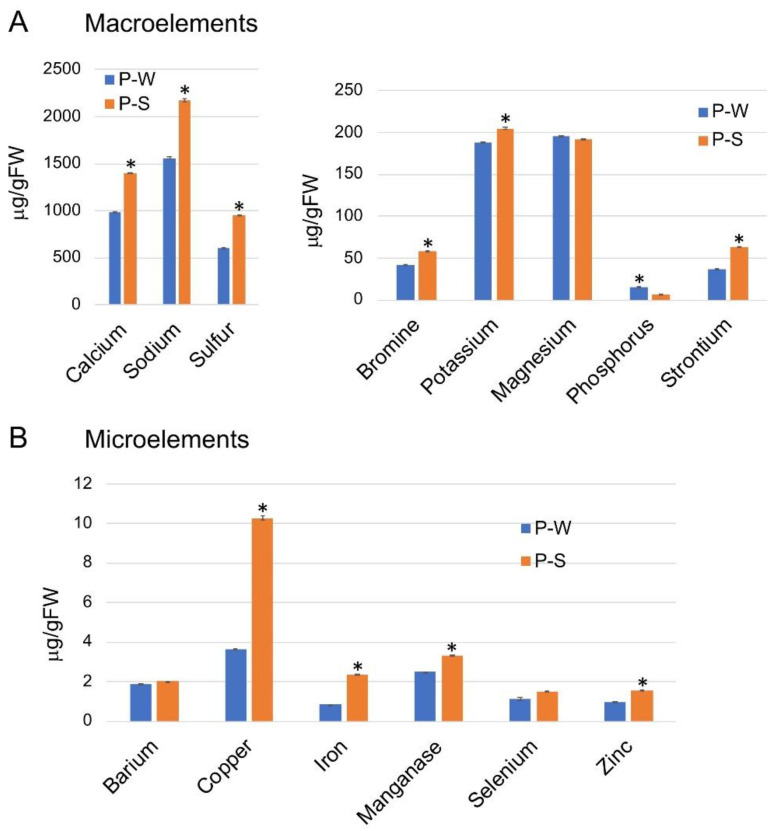
Nutrient profiles of *Z. dumosum* petioles during the winter (P-W) and the summer (P-S) months. Petioles extracts were subjected to ICP-OES (4 repeats) to detect (**A**) macroelements and (**B**) microelements. Vertical bars represent the standard deviation. Asterisks indicate statistically significant differences in element concentration between winter and summer petioles (*p* < 0.05; Student’s unpaired *t*-test).

**Table 1 plants-11-00484-t001:** Endophytic bacteria isolated from *Z. dumosum* petioles. Identity code of each endophyte is given in square brackets. Identity was based on blast analysis of their corresponding 16S rRNA encoding gene.

Strains		Identity %	Accession No	Source	Ref.
Phylum	Species				
*Actinobacteria*	[J6] *Actinotalea ferrariae*	98.99	KT 021826.1	Iron mine	[31]
	[J42] *Arthrobacter agilis*	98.15	NR_026198.1	Rhizosphere and phyllosphere	[32,33]
	[J53] *Arthrobacter oryzae*	99.14	NR_041545.1	Paddy soil	[34]
	[J33] *Arthrobacter subterraneus*	99.07	NR_043546.1	Subsurface water	[35]
	[J43] *Dietzia lutea*	99.27	NR_116462.1	Rhizosphere	[19]
	[J45] *Georgenia satyanarayana*	99.93	NR_117051.1	Soda lake	[36]
	[J8] *Kocuria rosea*	99.64	NR_044871.1	Rhizosphere	[37]
	[J29] *Pseudokineococcus basanitobsidens*	98.35	NR_158070.1	Volcanic rock	[38]
	[J52] *Salinibacterium xinjiangense*	97.84	NR_043893.1	Glacier	[39]
*Firmicutes*	[J46] *Bacillus fermenti*	98.53	NR_163641.1	Indigo fermentation liquid	[40]
	[J23] *Bacillus frigoritolerans*	99.86	NR_117474.1	Rhizosphere	[41,42]
	[J14] *Bacillus jeotgali*	99.65	NR_025060.1	Fermented seafood	[43]
	[J18] *Bacillus licheniformis*	99.79	NR_118996.1	Soil, rhizosphere, seawater	[44]
	[J27] *Bacillus megaterium*	99.79	NR_112636.1	Soil, seawater, sediment, food, fish	[45]
	[J40] *Bacillus sinesaloumensis*	98.18	NR_147383.1	Stool	[46]
	[J37] *Bacillus thioparans*	99.79	NR_043762.1	Subtropical estuary	[47]
	[J21] *Brevibacillus halotolerans*	99.72	NR_156834.1	Paddy soil	[48]
	[J3] *Domibacillus robiginosus*	97.91	NR_108861.1	Clean room	[49]
	[J25] *Fictibacillus nanhaiensis*	99.30	NR_117524.1	Bioreactor	[50]
	[J50] *Fictibacillus phosphorivorans*	98.78	NR_118455.1	Bioreactor	[50]
	[J56] *Paenisporosarcina indica*	98.88	NR_108473.1	Glacier	[51]

## Data Availability

The data that support the findings of this study are available in the main text and in the Appendix A.

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
