# Peer review of "Endophytic Bacteria Colonizing the Petiole of the Desert Plant Zygophyllum dumosum Boiss: Possible Role in Mitigating Stress"

_plants, 2022, doi:10.3390/plants11040484_

Round 1

Reviewer 1 Report

To me the manuscript is original, well written, easy to read and the conclusions are consistent.

I indicated minor revision because I found the authors missed the specification of which elements (not nutrients) determined, and, on the other hand, they must explain why determined elements like barium, strontium and so on( line 280).

The author must introduce some changes in the manuscript and for me this is a “minor revision”

Besides,  I found an apparent mistake the authors must corroborate. The primer cited by de Lillo et al., 2006 is not 1292 but 1492 (line 116).

Best regards.

Raúl

Author Response

Thanks

Author Response

Thanks for the thoughtful comments

Reviewer 3 Report

The isolation of endophytic microorganisms from various plant objects and the evaluation of their biological properties is one of the actively developing modern trends in biological science. The article by Srinivasan J. et al. “Endophytic bacteria colonizing the petiole of the desert plant Zygophyllum dumosum Boiss: possible role in mitigating stress” is devoted to the isolation of endophytic bacteria from Z. dumosum plants, the evaluation both of low molecular weight metabolites produced by them, and the effect of plant extracts, obtained in different seasons (summer and winter), on the growth characteristics of endophytes. On the basis of the obtained works, the authors of the work conclude that endophytes isolated from Z. dumosum plants contribute to the formation of plant resistance to extreme environmental conditions. The article is written quite clearly, but in the course of viewing the work, I had some comments on a number of positions:

  1. In the Materials and Methods section, it is noted (lines 92-94) that "Petiole samples for endophytes isolation were collected from Z.dumosum 92 shrubs growing in the Central Negev Highlands (30o51.12'01''N 34o 46.06'.42' 'E, at 511 m ele- 93 vation) on February 2020 from 12 plants within the 3-7 m distance between them". But it goes on to note (lines 97-98) that "For preparation of petiole extracts (PEs) we used petioles that were collected during 97 the summer (PE-S) and winter (PE-W) of 2010 and kept frozen at -80ºC until used". The question arises. Why the authors of the article could not use plant extracts isolated in 2020, but turned to those that were stored for 10 years to assess the growth characteristics of bacteria and other studies.
  2. It is noted that in the course of experiment, the authors isolated 76 morphologically different isolates from plant tissues of Z. dumosum, of which 60 were genetically characterized and a phylogenetic tree was constructed (Fig. 2). The authors further indicate that the isolates were divided into 21 clusters, which were identified as microbial species. At the same time, one must understand that each isolate is unique, and according to the rules, each of them must be assigned a number, and if there is already a complete characteristic, then it must be deposited in some kind of database. However, in the article, the authors in Table 1 show only a comparative characteristic of the isolates by 16S RNA. I consider it necessary for the authors to number their isolates and indicate in the text where they were deposited and subsequently stored.
  3. In fig. 3 shows that isolated endophytes (for example, the authors of the work took 6 isolates out of 21 characterized (why?)) actively react to the addition of an extract from Z. umosum plants to the nutrient medium. Indeed, the bacteria reacted positively to the plant extract. But I think that there are not enough strains to prove that it is the endophytes isolated from Z. umosum that react to the host extracts, for example, strains that are not endophytes for this plant. How do such strains (even E. coli, and preferably an endophyte not associated with Z. umosum) react to a plant extract. For example, I do not exclude that in a nutrient medium, a plant extract additionally introduces certain nutrients, which contribute to the active growth of microorganisms. Only and everything. Moreover, I did not find a significant difference between the extracts squeezed out of plants in summer and winter.
  4. Since the authors discuss the issue of the ability of bacteria to impart tolerance to plants, which is proved by the ability of bacteria under the influence of plant extracts to produce various metabolites (Fig. 4 and Fig. 5 (why in 2 figures)), including osmoprotectants. But the question arises as to how the bacteria themselves react to salt stress. Does the titer, as well as the species structure of bacteria in plants, change under the influence of stress. This is practically not discussed in the work.
  5. From fig. 5 it is not clear in what measurements (g/gFW) the content of metabolites in the culture was estimated
  6. I think that section “Z. dumosum petioles accumulate high levels of salt during the dry season” is redundant in this paper, as it is not discussed in relation to the endophytic microbiome.
  7. I think that the article needs serious revision and after the next stage of peer review, it can be published in the journal Plants

Author Response

Thanks for the thoughtful comments

Round 2

Reviewer 2 Report

Dear Autor,

thank you for handling my questions and criticisms.

  • I am still unhappy with the phrase in line 128 "different growth". What is meant by different?
  • Please mention that you did not find any fungi. Otherwise the readers will wonder.
  • Selection of media, it is not uncommon that not all microorganisms could be cultured. However, I expect a critical statement about this in the text.
  • Line 315 I agree with you that bacterial pigments play an often underestimated role. I also believe that it changes the metabolism of plants when bacteria grow here and release pigments. You refer here to the statement from Mohammadi et al. 2012 "visible as a bright yellow bacterial ooze and aids the cells in coping with UV light exposure encountered on the leaf surface."I think. I agree with that. Where I am critical is if this is supposed to be caused by natural endophytes. Pantoea is, after all, a pathogen. A plant's population of natural endophytes is usually much less concentrated than a pathogen. I would like to see a more critical statement in the text. Question out of curiosity, have you ever found color altered petioles?

Author Response

thank you for handling my questions and criticisms.

  1. I am still unhappy with the phrase in line 128 "different growth". What is meant by different?

This refers to the selection of endophytes for metabolic analysis. We indicated “different mode of growth”. As can be seen in Fig. 3, B. frigoritolerans demonstrates logarithmic phase of growth between 6 to 12 h and then slightly reduced, K. rosea showing log phase between 6 to 18h and then slightly increase, while A. agilis is showing a biphasic type of growth.

  1. Please mention that you did not find any fungi. Otherwise the readers will wonder.

Response: This is now clearly mentioned in the revised manuscript (Lines 222-223).

  1. Selection of media, it is not uncommon that not all microorganisms could be cultured. However, I expect a critical statement about this in the text.

Response: See lines 222-223.

  1. Line 315 I agree with you that bacterial pigments play an often underestimated role. I also believe that it changes the metabolism of plants when bacteria grow here and release pigments. You refer here to the statement from Mohammadi et al. 2012 "visible as a bright yellow bacterial ooze and aids the cells in coping with UV light exposure encountered on the leaf surface."I think. I agree with that. Where I am critical is if this is supposed to be caused by natural endophytes. Pantoea is, after all, a pathogen. A plant's population of natural endophytes is usually much less concentrated than a pathogen. I would like to see a more critical statement in the text. Question out of curiosity, have you ever found color altered petioles?

We brought here Mohammadi et al article to show that pigments produced by microorganisms associated with plants could have a protection role including against UV light.  As you have mentioned, it is likely that “A plant's population of natural endophytes is usually much less concentrated than a pathogen” but we cannot exclude the possibility that in spite of low concentration they might have local protective effect on cellular components.

Regarding the question: “have you ever found color altered petioles?”

We cannot answer this question, as we never addressed this aspect of petiole color.

Reviewer 3 Report

The authors of the article made some minor changes to the article. I understand that the obtained isolates are retained by the authors.
But the most important issue of this article was related to the fact that in the work we are studying not species with endophyticity, but isolates belonging to one or another species. This important note has not been corrected. The number of the isolate must be indicated on the figures. The reader may realize that all strains of an identified species are endophytic. Let the authors provide information about the isolates isolated and identified by them. In the work cited at this time, this information is not available and we see only an approximate species structure of microorganisms isolated by the authors.

Author Response

The authors of the article made some minor changes to the article. I understand that the obtained isolates are retained by the authors.
But the most important issue of this article was related to the fact that in the work we are studying not species with endophyticity, but isolates belonging to one or another species. This important note has not been corrected. The number of the isolate must be indicated on the figures. The reader may realize that all strains of an identified species are endophytic. Let the authors provide information about the isolates isolated and identified by them. In the work cited at this time, this information is not available and we see only an approximate species structure of microorganisms isolated by the authors.

Response: We hope we understood the comment. Thus, although we included the code number of each isolate in Table 1, now we also indicated the isolate number in the figures when appropriate.